# Invertebrate Models Untangle the Mechanism of Neurodegeneration in Parkinson’s Disease

**DOI:** 10.3390/cells10020407

**Published:** 2021-02-16

**Authors:** Andrei Surguchov

**Affiliations:** Department of Neurology, University of Kansas Medical Center, Kansas City, KS 66160, USA; asurguchov@kumc.edu; Tel.: +1-(913)-689-0771

**Keywords:** neurodegeneration, protein aggregation, protein sorting, alpha-synuclein, vesicular trafficking, folate, epigenetic regulation, DNA methylation

## Abstract

Parkinson’s disease (PD) is the second most common neurodegenerative disease, afflicting ~10 million people worldwide. Although several genes linked to PD are currently identified, PD remains primarily an idiopathic disorder. Neuronal protein α-synuclein is a major player in disease progression of both genetic and idiopathic forms of PD. However, it cannot alone explain underlying pathological processes. Recent studies demonstrate that many other risk factors can accelerate or further worsen brain dysfunction in PD patients. Several PD models, including non-mammalian eukaryotic organisms, have been developed to identify and characterize these factors. This review discusses recent findings in three PD model organisms, i.e., yeast, Drosophila, and *Caenorhabditis elegans*, that opened new mechanisms and identified novel contributors to this disorder. These non-mammalian models share many conserved molecular pathways and cellular processes with humans. New players affecting PD pathogenesis include previously unknown genes/proteins, novel signaling pathways, and low molecular weight substances. These findings might respond to the urgent need to discover novel drug targets for PD treatment and new biomarkers for early diagnostics of this disease. Since the study of neurodegeneration using simple eukaryotic organisms brought a huge amount of information, we include only the most recent or the most important relevant data.

## 1. Introduction

The aging of the human population is associated with many new challenges requiring urgent solutions. One of such challenges is an increase in the number of people afflicted by age-related neurodegenerative disorders. Among these disorders, the fastest growing is Parkinson’s disease (PD), the most prevalent movement disorder and the second most prevalent neurodegenerative disease. PD growth is surpassing that of currently the most prevalent, Alzheimer’s disease [1,2]. During the 25 years from 1990 to 2015, the prevalence of PD, as well as disability and mortality due to this disorder, more than doubled [1]. PD affects around 1 million individuals in the USA alone, and the number of people with PD may increase from 6.9 million in 2015 to 12.9–14.2 million in 2040 [2], creating tremendous socioeconomic challenges.

Now, in 2021 we face a unique condition when two pandemics, COVID-19 and PD met [3]. We may consider the threat of COVID-19 with more optimism since several vaccines developed against the virus will hopefully reduce the prevalence and negative consequences of this infection. Unfortunately, for PD and other neurodegenerative diseases, there are no such straightforward preventive measures, and therefore, they require a comprehensive and thorough investigation of their molecular mechanisms.

The global burden of PD and other neurodegenerative diseases emphasizes the critical requirement for inventive strategies to define new drug targets and disease-modifying factors.

## 2. PD Pathogenesis

PD is the most common complex neurodegenerative movement disorder, with unclear etiology in the majority of patients. The pathogenesis of PD is influenced by a complex and mostly elusive interplay between genetic and environmental factors. A hallmark of PD is the accumulation of eosinophilic cytoplasmic inclusion—Lewy body and Lewy neurites in neurons and glia [4]. α-Synuclein, a protein highly expressed in the CNS, has been identified as the main component of the insoluble filaments forming the Lewy body [5].

Other factors, such as mitochondrial dysfunction, alterations in dopamine production and metabolism, and impairment of protein degradation pathways also contribute significantly to PD pathogenesis [6]. Discovery of several mutations in α-synuclein genes [7,8,9] and gene duplication and triplication [10,11,12,13] allowed an unveiling of the basic molecular mechanisms of this disorder but failed to reveal the contribution of loci with lower penetrance in PD pathogenesis. PD is primarily idiopathic, while genetic forms of PD are rare. Mutations causing this disease are found in approximately 2% of cases [14]. According to other estimates, about 10% of PD cases are considered to involve genetic factors [15,16]. Therefore, in the majority of patients, the disease develops as a result of the combination of mutations in multiple PD-associated genes and environmental risk factors [17]. More than twenty years of genetic research in PD have led to the identification of several monogenic forms of the disorder and numerous genetic factors increasing the risk of developing this disease.

A meta-analysis of genome-wide association studies (GWAS) identified more than one hundred loci affecting PD progression [18]. Some of the proteins encoded by these loci impact the disease’s course, increasing α-synuclein aggregation potential. Others regulate α-synuclein toxicity, but the exact mechanism of their action remains elusive for most of them.

It becomes clear that identifying components and pathways mediating α-synuclein neurotoxicity and finding other players implicated in PD pathogenesis is crucial for developing of drugs and finding new diagnostic tools for early detection of this disorder. Models of PD in non-mammalian eukaryotic organisms play a notable role in untangling complex interactions between factors implicated in PD pathogenesis.

## 3. Simple Eukaryotic Organisms as Models of PD

Simple non-mammalian model organisms, i.e., the baker’s yeast *Saccharomyces cerevisiae (S. cerevisiae)*, the nematode worm *Caenorhabditis elegans (C. elegans)*, and the fruit fly *Drosophila melanogaster* (*D. melanogaster*), have been successfully used to study both environmental and genetic factors and provide insights to the pathways relevant for PD pathogenesis. These model organisms can breed in large numbers, have a short generation time, and possess genes similar to humans that can be used to create highly detailed genetic maps. It is important to note that yeast, *C. elegans*, and Drosophila do not express α-synuclein—an important player in PD pathology [19]—so experiments were performed in an α-synuclein null genetic background.

### Advantages of Non-Mammalian Eukaryotic Models of PD

Haploid budding yeast cell contains a tiny genome built of only 12 megabases of genomic DNA and 5885 potential protein-encoding genes [20,21]. This model organism allows using a tremendous power of yeast genetics.

The adult *C. elegans* hermaphrodite form has only 959 somatic cells, but it has various organs and tissues, including the nervous system consisting of about only 300 neurons and 56 glial cells [22]. The *C. elegans* genome shares greater than 83% homology with the human genome [23]. The main functional components of mammalian synaptic transmission, such as neurotransmitters, ion channels, receptors, and transporters, are conserved in *C. elegans*. Furthermore, the comparison of the human and *C. elegans* genomes demonstrated that most of the human disease genes and disease pathways are present in this model organism [24].

Drosophila is well-investigated at the phenotype level and has a relatively simple genome with approximately 75% of functional homologs of human disease-causing genes [25]. This simple eukaryotic organism has various cellular processes relevant to all eukaryotes, including humans. Therefore, it is a powerful model organism for studying fundamental aspects of eukaryotic cell biology.

## 4. Yeast as a Convenient Model to Study PD Pathogenesis

The complexities of PD cannot be precisely mirrored in a single-cell model because a human disease includes a variety of autonomous processes leading to systems-level dysfunction. However, expression of α-synuclein in yeast, which lacks synuclein homologs, recapitulates some properties of its cellular behavior in human cells, such as aggregation, membrane binding, and toxicity [26]. The results of genetics and biochemical experiments with *S. cerevisiae* have contributed considerably to understanding the basic biology of various cellular processes, such as membrane trafficking, signal transduction, and protein turnover.

A substantial homology of the yeast genome with humans enables the study of human gene functions in yeast using genetic complementation. As a result, yeast became a convenient model to examine cell death and survival mechanisms, the secretory pathway, and protein quality control systems [27,28].

### 4.1. α-Synuclein Overexpression in Yeast

Overexpression of α-synuclein (gene structure is shown in Figure 1A) affects vesicular trafficking, impairs proteasomal activity, and causes disturbances in lipid metabolism [26,27,28].

Later well-designed experiments revealed mitochondrial dysfunction as a consequence of α-synuclein toxicity [29]. A similar approach demonstrated an association of the voltage-dependent anion channel mitochondrial and endonuclease G with α-synuclein (reviewed in [29]). Several pathways have been linked to α-synuclein toxicity by a similar approach, including oxidative stress, autophagy and mitophagy, trafficking between the endoplasmic reticulum (ER) and Golgi, and endocytosis [28,30,31].

### 4.2. α-Synuclein Blocks ER-to-Golgi Vesicular Trafficking

Cooper and coauthors [31] developed a regulated α-synuclein expression in yeast with increased gene dosage using a galactose inducible promoter. They found that overexpression of human α-synuclein caused ER stress, blocked ER-to-Golgi vesicular trafficking, and caused accumulation of misfolded α-synuclein. The results of the genome-wide screen revealed that Rab guanosine triphosphatase Ypt1p is associated with α-synuclein inclusions. Importantly, the authors further employed multiple model systems to support the yeast model’s first results.

Detailed investigation using Drosophila, *C. elegans*, and primary rat neuronal cultures confirmed that increased expression of a key vesicular trafficking pathway protein, Rab1, could rescue dopaminergic neuron loss induced by α-synuclein [31]. Thus, the results in cellular models revealed links between multiple Parkinsonian risk factors and biological processes, demonstrating that accumulation of misfolded α-synuclein might disrupt basic cellular functions, including ER–Golgi trafficking. This significant breakthrough in understanding the mechanism of neurodegeneration using yeast cells was made by a group of researchers led by Susan Lindquist. She and her colleagues showed how powerful are yeast-to-human approaches [27,28,30]. These initial findings demonstrate how powerful integrating yeast genetics, genomics, and computational biology have paved the way for new studies in yeast to identify novel components implicated in PD pathogenesis.

### 4.3. Protein Sorting and Degradation

New components in protein sorting and degradation affecting PD pathogenesis were identified in the yeast model of PD. For example, in a set of elegant experiments, Aufschnaiter and coauthors [32,33] revealed an important role of cathepsin D and Ca^2+^/calmodulin-dependent phosphatase calcineurin in endosomal sorting and lysosome degradation of α-synuclein and other aggregation-prone proteins. Cathepsin D is a lysosomal aspartyl protease highly expressed in the brain and critically involved in the degradation of unfolded or damaged proteins delivered into lysosomes via endocytosis or autophagy. Experiments in yeast demonstrated that the cytoprotective effect of cathepsin required functional calcineurin signaling. Active calcineurin was essential for the appropriate trafficking of cathepsin D to the lysosome and for recycling its endosomal sorting receptor to permit the next rounds of shuttling.

### 4.4. Role of Myosin in PD Pathogenesis

More recently, Nguyen and coauthors [34] identified a specific type of myosin motors, i.e., *myo2,* implicated in the retrograde traffic of proteins from the endosome toward the trans-Golgi network [34]. Myo2 is one of several types of myosin in yeast cells responsible for intracellular motility. Its function is to serve as a motor to transport various cargoes on actin filaments. Myo2 has actin-binding and cargo-binding domains. Mutations in either of these domains cause severe defects in sorting and trafficking at the endosome.

This process is a key intracellular step regulating cellular homeostasis, and its dysfunction is associated with the pathogenesis of PD. Using the yeast model, the authors found that both the N-terminal actin-binding domain and the C-terminal tail domain of Myo2 are required for optimal cargo protein traffic. Myo2 can interact with vacuole-derived vesicles or endosome and walks along the actin cable toward the Golgi. Near Golgi, the positive end of actin cable concentrates for effective docking of transported membrane cargo to a cargo entry site [34].

Another important finding of PD pathogenesis discovered in the yeast model is the effect of α-synuclein on lipid metabolism. Expression of α-synuclein in yeast significantly changed lipidomic profiling increasing oleic acid (OA, 18:1), triglycerides, and diglycerides. These results demonstrate that monounsaturated fatty acid metabolism is essential for α-synuclein associated neurotoxicity [35]. This finding is important because lipid and fatty acid homeostasis are vital factors in neurotransmission and receptor activation.

Thus, the studies in *S. cerevisiae* offered enormous potential for a better understanding of human diseases, basically due to a short cell cycle, small genome, cell organization similarity, and advanced genetics.

However, an evident disadvantage of the yeast system for investigating of the human brain is the absence of multicellular organization, CNS, and synaptic connection between cells.

Two other model organisms—*C. elegans* and *D. melanogaster*—opened an approach for the exploration of more complex functions specific for multicellular organisms.

## 5. Studies with Nematode *C. elegans* Brought New Findings of PD Pathogenesis

The main functional constituents of mammalian synaptic transmission, such as receptors, neurotransmitters, transporters, and ion channels, are preserved in *C. elegans*. This model organism has orthologs of many PARK genes implicated in PD, with the notable exception of PARK1/SNCA (α- synuclein). This feature conveniently allows *C. elegans* researchers to overexpress α-synuclein without interference from endogenous α-synuclein or a dominant-negative effect since the worm essentially serves as an α-synuclein null genetic background. Recent findings in *C. elegans* revealed the role of several new proteins in PD pathology.

### 5.1. A Link between Dopamine- and α-Synuclein-Mediated Toxicities

Mor and coauthors [36] used *C. elegans* to investigate an association between dopamine and α-synuclein-mediated toxicities. More specifically, the researchers wanted to reveal whether dopamine could modify α-synuclein oligomerization in vivo and whether the resultant species are capable of driving neurodegeneration. The authors changed α-synuclein amino acid sequence Y125EMPS129 at the site of interaction with dopamine located in the C terminus to F125AAFA129. α-Synuclein mutated at the site of the interaction with dopamine prevented dopamine-induced toxicity. Thus, this neurotransmitter’s toxicity was abolished without its interaction with α-synuclein, and dopaminergic neurons could be rescued from dopamine toxicity by inhibiting its interaction with α-synuclein. These findings point to the importance of the interaction of dopamine and α-synuclein in the neurodegenerative process.

### 5.2. PARK9 (ATP13A2) in PD Pathogenesis

Recent studies in *C. elegans* revealed that defects in ATP13A2 might also contribute to PD pathogenesis. ATP13A2 is a late endolysosomal transporter that carries the polyamines spermidine and spermine from the late endo/lysosome to the cytosol [37]. ATP13A2-mediated polyamine transport may alleviate oxidative stress and thus preserve mitochondrial health. On the contrary, ATP13A2 deficiency or dysfunction disrupts lysosomal polyamine export and causes accumulation of polyamines in lysosomes, which may lead to its rupture. Lysosomal polyamine toxicity is one of the factors causing neurodegeneration.

Moreover, ATP13A2 protects against rotenone and mitochondrial-generated superoxide (MitoROS). ATP13A2 knockdown induces a stress response after exposure to rotenone. Studies in *C. elegans* identified a conserved cell-protective pathway that alleviates mitochondrial oxidative stress via ATP13A2-mediated lysosomal spermine export [38].

### 5.3. DP-1/TDP-43 Potentiates Human α-Synuclein Neurodegeneration in C. elegans

The *C. elegans* model of PD was also used to investigate the interaction of α-synuclein with a DNA/RNA binding protein called TDP-43 [39]. Aggregates of TDP-43 are colocalized with α-synuclein in Lewy body; however, the details of their interaction were unclear. The researchers studied the effect of genetic deletion of tdp-1, the *C. elegans* ortholog of human TDP-43, in transgenic *C. elegans* overexpressing wild type and A53T mutant α-synuclein. The *C. elegans* tdp-1 enhances the expression of lysosomal genes and reduces the activity of genes involved in heat shock response. The authors assume that TDP-43 potentiates the neurotoxicity of α-synuclein in *C. elegans,* affecting gene expression of specific biochemical pathways. TDP-43 and α-synuclein may be considered as cooperative mediators of neurodegeneration.

The results of this study demonstrate that TDP43 interacts with α-synuclein at several points, including regulation at mRNA and protein levels, control of expression of heat-shock protein genes, and potentiation of α-synuclein aggregation. Thus, TDP-43 acts through multiple pathways to stimulate the neuropathology of α-synuclein.

### 5.4. α-Synuclein Self-Assembly and Lewy Body Formation

In a recent study, Hardenberg and coauthors [40] described how the *C. elegans* model of PD helped to understand the relationship between α-synuclein self-assembly and the Lewy body formation. The authors tested a hypothesis that α-synuclein might be capable of irreversibly capturing cellular components through liquid–liquid phase separation. A similar mechanism has been demonstrated to drive the self-assembly of several disease-associated proteins on a pathway to the establishment of solid aggregates.

The authors found that α-synuclein underwent a two-step transition. On the first step, α-synuclein passes through liquid‒liquid phase separation by forming a liquid droplet state. On the second step, this liquid droplet state is converted into an amyloid-rich hydrogel possessing Lewy-body-like properties. Interestingly, the maturation process toward the amyloid state may be hindered in vitro in the presence of model synaptic vesicles. This data allows proposing that the formation of the Lewy body is associated with the blocked maturation of α-synuclein condensates in the presence of lipids and possibly other intracellular substances. Cellular α-synuclein droplets may be converted into perinuclear aggresomes in a process regulated by microtubules [41].

## 6. New Discoveries in Drosophila

*D. melanogaster* is a classic model system offering plenty of genetic tools that enable investigations of biochemistry using both forward and reverse genetics. Drosophila models have provided detailed insights into the pathogenic mechanisms of PD [42].

### 6.1. Role of SPEN/SHARP Genes

SPEN/SHARP is one of many astrocyte-expressed genes that are differentially expressed in the substantia nigra of PD patients compared with control subjects [43]. Interestingly, the differentially expressed genes are enriched in lipid metabolism-associated genes. In a Drosophila model of PD, flies carrying a loss-of-function allele of the ortholog split-ends (spen) or with glial cell-specific, but not neuronal-specific, spen knockdown were more sensitive to paraquat intoxication, indicating a protective role for spen in glial cells. Spen is a positive regulator of Notch signaling in adult Drosophila glial cells. Moreover, spen is required to limit the abnormal accumulation of lipid droplets in glial cells in a manner independent of its regulation of Notch signaling. These results demonstrate that spen regulates lipid metabolism and storage in glial cells and contributes to glial cell-mediated neuroprotection.

### 6.2. RING Finger Protein 11 (RNF11)

RNF11 is a modulator of protein degradation and master regulator of a plethora of signaling pathways. It modulates ubiquitin-regulated processes that are involved in transforming growth factor beta (TGF-β), nuclear factor-κB (NF-κB), and epidermal growth factor (EGF) signaling pathways. Privman-Champaloux and coauthors [44], using a Drosophila model, demonstrated that RNF11 affected both dopamine release and uptake. RNF11 expression decreased in human dopaminergic neurons during PD. This reduction is presumably protective by increasing dopamine neurotransmission in the surviving dopaminergic neurons.

### 6.3. Folate Metabolism and α-Synuclein Toxicity

Several studies in Drosophila pointed to a folate metabolism as an important mediator of α-synuclein toxicity. Recently, Sarkar and coworkers [45] conducted a gene ontology enrichment analysis and examined proteomic changes relevant to PD in a Drosophila model. Their results indicate that folate metabolism and GTP cyclohydrolase (GCH1) play an essential role in regulating α-synuclein toxicity. GCH1 is a member of the folate and biopterin biosynthesis pathways catalyzing guanosine triphosphate (GTP) hydrolysis and its conversion to 7,8-dihydroneopterin triphosphate. Tetrahydrobiopterin is a cofactor in the biosynthesis of several monoamine neurotransmitters, including dopamine.

GCH1 and folate metabolism can mediate α-synuclein toxicity via several mechanisms, including their role in neurotransmitter production and abnormal dopamine metabolism, involvement in energy production pathway, including regulation of mitochondrial function, and oxidative stress. Furthermore, folate may serve as a donor of methyl groups, which can be used, in particular, for DNA methylation—the most studied epigenetic modification (Figure 1B,C) [46,47].

### 6.4. Folate and Methyl Group Production

Folate is one of the vital cofactors involved in DNA methylation modification [48]. The intracellular level of α-synuclein—one of the important players in PD pathogenesis (see Chapter 1)—is regulated on several levels, including by epigenetic mechanism. The term epigenetics refers to reversible adjustments in gene expression, which can be inherited but are not engraved in the DNA sequence. These alterations can be applied via various mechanisms, one of which is methylation of the DNA. DNA methylation is one of the important steps involved in the deregulation of α-synuclein expression in PD [49].

### 6.5. Folate and Folic Acid: Biochemistry and Physiology

Folate (vitamin B9 or folacin) is a water-soluble vitamin belonging to the B-complex group of vitamins. Folate is a generic name for a group of related compounds with similar properties. The active form of vitamin B9 is folate, known as levomefolic acid or 5-methyltetrahydrofolate (5-MTHF). There is a difference between the terms folate and folic acid (FA), since folate is the natural form of vitamin B9 in food, whereas FA is a synthetic form. Nevertheless, their names are frequently used interchangeably. Folic acid is more stable during processing and storage and is easily converted into folate by the body. In the digestive system, most dietary folate is converted into the biologically active form of vitamin B9-5-MTHF before entering the bloodstream [50].

## 7. Conclusions

Many essential molecular and cellular mechanisms implicated in PD have been discovered in small invertebrate organisms, predominately in yeast, fly, and nematode. Studies in these model organisms unveiled several important mechanisms and identified new players involved in PD’s pathogenesis. Finding these new components opens new insights into the disease pathology and offers approaches for further exploration of them as targets for therapy. An example of this approach’s efficiency in potential drug testing was recently proven in a study of doxycycline as a potential treatment for PD. Dominguez-Meijide and coauthors [51] used a *C. elegans* model of PD and found that doxycycline induced a cellular redistribution of α-synuclein aggregates and decreased their number and size. Furthermore, this antibiotic lowered the production of reactive oxygen species by mitochondria. These results demonstrate that doxycycline treatment [51] may be an effective strategy against PD and other synucleinopathies [52,53]. A wide range of approaches were applied by molecular, genetic, and chemical manipulations of gene function, i.e., using transgenic overexpression of exogenous human disease-related proteins, mutagenesis (transposon-based insertion, deletion libraries, etc.), or RNA interference (RNAi)-mediated knockdown to determine the loss- or gain-of-function phenotypes. Novel methods allow researchers to investigate the model organisms at the system level, ultimately permitting the reconstruction of many processes in silico. However, it should be stressed that genes shown to affect the pathogenic pathway in one model do not always have a similar effect in other models, perhaps because of the intrinsic dissimilarities between species or differences between the methods employed. Therefore, to establish the value of genes discovered in small model organisms, the results should be reproduced and validated in mammalian animal models and human neuronal cell cultures. If the function of new components identified in small organisms is evolutionarily conserved, their mammalian counterparts may become therapeutic targets for further pharmacological examination. Moreover, small non-mammal model organisms can be used to understand the basic mechanisms underlying the causal gene of disease and also as a pharmacological screening tool.

The model organisms have undoubtedly been very useful in identifying particular genes or pathways involved in human diseases. They helped scientists to collect a huge amount of knowledge. However, model systems should not be considered as ideal platforms for research in neurodegeneration and other fields. Several unsuccessful drug trials against human diseases derived from studies with these models stressed that no model organism is expected to truly replicate the human disease phenotype [54,55]. The development of new methods established on in silico and stem cell-based techniques open alternative pathways for the investigation.

## Figures and Tables

**Figure 1 cells-10-00407-f001:**
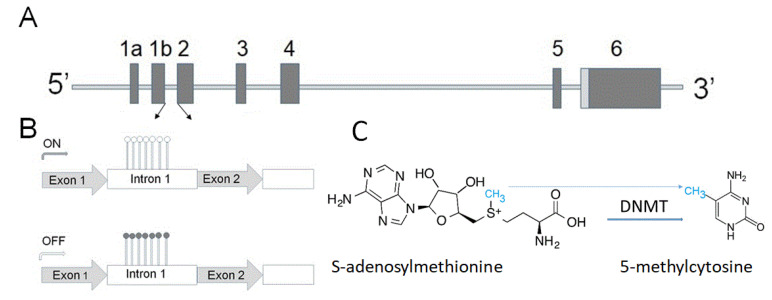
(**A**) Human α-synuclein gene contains six exons shown as vertical bars. The translation start codon ATG is located in exon 2. The sequence between two arrows contains a high density of CpG bases, which can be methylated (**B**,**C**). (**B**) Upper panel shows a small level of methylation (open circles) within the α-synuclein intron 1 region, allowing a high level of the gene expression (ON). Lower panel-methylation of cytosine residues (closed circles) downregulates expression of the α-synuclein gene. (**C**) DNA methylation catalyzed by methyltransferases (DNMTs) converts cytosine to 5-methylcytosine by transfer of methyl group from S-adenosylmethionine. This is one of several epigenetic mechanisms to control gene expression.

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
