# Peer review of "Invertebrate Models Untangle the Mechanism of Neurodegeneration in Parkinson’s Disease"

_cells, 2021, doi:10.3390/cells10020407_

Round 1
Reviewer 1 Report
The review entitled Non-mammalian eukaryotic models untangle the mechanism of neurodegeneration describes aspects of non-verterbrate models of Parkinson's disease (PD). Whereas the topic of the manuscript is of high interest to the broad readership of the journal, the content is mostly superficial and needs thorough revision before being suitable for publication.
Major comments:
1.) This manuscript lacks a unique selling point. What is the difference to the numerous other reviews on this topic?
2.) The specific content of the manuscript does not fit to the very general title of the paper. The paper focuses on PD and not on neurodegeneration in general. Important non-mammalian model systems, like zebrafish, are missing. Title and content must fit together!
3.) The content on yeast models of PD is very superficial, and the examples shown in this paragraph are selected rather randomly (or at least I cannot find any concept here). There are numerous yeast PD models affecting numerous cellular pathways. This is not reflected in this manuscript at all. Similar is true for the chapters with Drosophila and C. elegans.
4.) The selected figures are either meaningless (Figure 1) or very specific (Figure 2) and do not support the general claim of the review. Both figures are of poor quality.
Author Response
Many thanks for reviewer’s critique, comments and suggestions. They helped us to improve the quality of our manuscript. Below are point-by-point details of the revisions and answers to reviewer’s criticism:
Reviewer 1
1.) This manuscript lacks a unique selling point. What is the difference to the numerous other reviews on this topic?
Response. A focus of this manuscript is on new publications (14 references in the Bibliography are published in 2020/2021 and three at the end of 2019, so we included in the review the data from 17 recent articles. We paid special attention to the latest articles that identified novel players in Parkinson’s disease (PD) or new processes related to PD pathology. A few previous reviews (if any) are focused on PD with discussion of novel findings made in 2020-21 using invertebrate models.
Examples:
- a) Publication by van Veen and coauthors in Nature in 2020 (ref 35) shows in elegans model that ATP13A2 lysosomal polyamine exporter with the highest affinity for spermine. Furthermore, the authors proved that defective lysosomal polyamine export is a mechanism for lysosome-dependent cell death implicated in neurodegeneration.
- b) An article by Vrijsen and coauthors published in PNAS USA, 2020 revealed a new mechanism by which ATP13A2-mediated endo-lysosomal polyamine export countered mitochondrial oxidative stress and provided protection against mitochondrial toxins. Using elegans the authors showed that ATP13A2 effectively lowered ROS levels and promoted mitochondrial health and functionality.
- c) Hardenberg and coauthors (JMCB, 2021) (ref 38) used a elegans model of PD to reveal new initial steps in α-synuclein aggregation. The authors showed that α-synuclein underwent liquid‒liquid phase separation by forming a liquid droplet state.
- d) Ray and coauthors (ref 39) in an article published in Nature Chemistry (2020) continued the investigation of this process and discovered that α-synuclein droplets transform into perinuclear aggresomes in the process regulated by microtubules. This is a new mechanism is highly relevant to PD pathogenesis.
- e) An article published by Nguyen V et al. (ref 33, November 2020) contains a first prove of myosin V role in PD. These results received in a yeast model provided novel insights into the function of Myo-family proteins in the recycling traffic.
- f) Girard and coworkers (Sci Rep, 2020, ref. 41) using Drosophila model for the first time proved that RNA-binding protein SPEN/SHARP modulates lipid droplet content and protects against paraquat toxicity.
2.) The specific content of the manuscript does not fit to the very general title of the paper. The paper focuses on PD and not on neurodegeneration in general. Important non-mammalian model systems, like zebrafish, are missing. Title and content must fit together!
Response.
Thank you for this suggestion. To fit the title and content we changed the title of the manuscript to “Invertebrate models untangle the mechanism of neurodegeneration in Parkinson’s disease”
3.) The content on yeast models of PD is very superficial, and the examples shown in this paragraph are selected rather randomly (or at least I cannot find any concept here). This is not reflected in this manuscript at all. Similar is true for the chapters with Drosophila and C. elegans.
Response.
We agree that there are numerous yeast PD models affecting numerous cellular pathways. Our inclusion criteria for selection of articles was based on novelty and importance. As we replied in Response to critique 1.) in our manuscript we included the newest publications (2020-2021) and only briefly mentioned those describing the most important finding published before (for example, milestone findings by Lindquist group, ref 26, 27, 29, 30).
4.) The selected figures are either meaningless (Figure 1) or very specific (Figure 2) and do not support the general claim of the review. Both figures are of poor quality.
Response. In response to this comment, we deleted Figure 1 and modified Figure 2. Correspondingly, we changed the legend to Figure 1 C to “DNA methylation catalyzed by methyltransferases (DNMTs) converts cytosine to 5-methylcytosine. This is one of several epigenetic mechanisms to control gene expression”.

Reviewer 2 Report
A manuscript by Surguchov provide a comprehensive review of literature on non-mammalian eukaryotic models of neurodegeneration. This is a strong manuscript - well written.
Comments:
- The title of the manuscript is misleading. It promises to cover neurodegenration in general but the entire manuscript is dedicated to Parkinson disease (PD). So I suggest changing the general term neurodegeneration to Parkinson disease.
- Yeast, C.elegans and Drosophila don't express alpha-Synuclein. It will be good to make it clear in the introduction for novice readers. Stating this in no way should be seen as a pitfall (in fact this is stated elsewhere in the manuscript as a strength).
- The previous concern brings me to this point - limitations of the three models. No model is perfect. But it is important to being critical about models. So I suggest dedicating a paragraph to discuss the limitation of models relevant to this manuscript.
- Section 3-4: Role of myosin in PD pathogenesis: The cited reference #33 has a limited relevance for this manuscript. I do appreciate the role of myosin in PD pathogenesis (eg Zhang et al, PNAS 2020, Myosin 7b). The entire section based of non-exciting evidence related to PD may be avoided. Instead, I would suggest adding a section on lipid dyshomeostasis. There are several interesting articles from the Lindquist lab and others highlighting the role of lipid metabolism in PD.
Author Response
Thanks for your review.
According to the reviewer's suggestions to add some maaterisl to the manuscript, I added 2 new references to the bibliography
19 Surguchev AA, Surguchov A. Synucleins and Gene Expression: Ramblers in a Crowd or Cops Regulating Traffic? Front Mol Neurosci. 2017, 13; 10:224.
35 Fanning S, Haque A, Imberdis T, Baru V, et al., Lipidomic Analysis of α-Synuclein Neurotoxicity Identifies Stearoyl CoA Desaturase as a Target for Parkinson Treatment. Mol Cell. 2019; 73 (5):1001-1014.e8.
Reviewer 3 Report
Dear editor,
I read with great interest the review article titled “Non-mammalian eukaryotic models untangle the mechanism of neurodegeneration” by Andrei Surguchov. Overall, I find the manuscript interesting, well-written and it revisits the recent developments in PD research with yeast, C. elegans and Drosophila melanogaster.
My main concern is related to the advocation of these model systems as ideal platforms for research in neurodegeneration. I want to note that my intentions are not to create tensions regarding which system is better than others, every model system has advantages and disadvantages, but it should be noted that all the systems discussed in this article are evolutionary greatly distant from the humans in several aspects. Moreover, considering the findings over the last years and the numerous unsuccessful drug trials against neurodegenerative diseases derived from studies with in vivo models, I think there is a wide scepticism already with more advanced and complicated rodent systems, so I think a similar indication should be noted for these non-mammalian systems. Finally, while the majority of genes implicated in PD and AD have relevant orthologues in these systems, for many cases the function and the number of isoforms is not exactly reflected and therefore it is not clear how the associated cellular processes might be different.
Additionally, I do not think that the manuscript benefits from the two figures. Especially the first one is a very generic figure that can be found in many other similar reviews and it might be better to remove it. Regarding figure 2, I think the schematic of the a-syn gene in panel A is useful and the panel B could be interesting especially linked with the text about the regulation of gene expression, but as it stands now it is not apparent to the reader what exactly these circles mean. Panels C and D from this figure do not contribute to the manuscript and in my opinion, they should be removed.
Finally, I have two minor notes:
Line 134: “Myo2 belongs to class-V myosins, which serve as motors that function as vehicles transporting intracellular cargoes via actin filament-dependent motility” I am not sure I understand what this sentence is talking about.
Lines 188-200: the author simply repeats the same things again and again in these lines and I cannot really see how the length of this part contributes to the manuscript.
Author Response
Many thanks for reviewer’s critique, comments and suggestions. They helped us to improve the quality of our manuscript. Below are point-by-point details of the revisions and answers to reviewer’s criticism:
1.) “…it should be noted that all the systems discussed in this article are evolutionary greatly distant from the humans in several aspects… My main concern is related to the advocation of these model systems as ideal platforms for research in neurodegeneration”
Response. We agree with this comments and added the following paragraph and 2 references to the end of the manuscript:
“The model organisms have undoubtedly been very useful in identifying particular genes or pathways involved in human diseases. They helped scientists to collect a huge amount of knowledge. However, model systems should not be considered as ideal platforms for research in neurodegeneration and other fields. Several unsuccessful drug trials against human diseases derived from studies with these models stressed that no model organism is expected to truly replicate the human disease phenotype (51, 52). The development of new methods established on in silico and stem-cell-based techniques open alternative pathways for the investigation”.
Added references:
- Marian AJ. Modeling human disease phenotype in model organisms: "It's only a model!". Circ Res. 2011;109(4):356-359.
- Hunter, Philip. “The paradox of model organisms. The use of model organisms in research will continue despite their shortcomings.” EMBO reports vol. 9,8 (2008): 717-20.
2.) Additionally, I do not think that the manuscript benefits from the two figures. Especially the first one is a very generic figure that can be found in many other similar reviews and it might be better to remove it. Regarding figure 2, I think the schematic of the a-syn gene in panel A is useful and the panel B could be interesting especially linked with the text about the regulation of gene expression, but as it stands now it is not apparent to the reader what exactly these circles mean. Panels C and D from this figure do not contribute to the manuscript and in my opinion, they should be removed.
Response.
As indicated in Figure legend (B), open circles designate a low level of methylation allowing a high level of gene expression (ON). Closed circles indicate high level cytosine methylation associated with a low level of a-synuclein expression (OFF).
As recommended, we deleted Figure 1 and panels C and D from Figure 2. These two panels we replaced with a more relevant panel C and added a corresponding legend for this panel.
3.) Finally, I have two minor notes:
Line 134: “Myo2 belongs to class-V myosins, which serve as motors that function as vehicles transporting intracellular cargoes via actin filament-dependent motility” I am not sure I understand what this sentence is talking about.
Response:
We corrected this sentence as follows:” Myo2 is one of several types of myosin in yeast cells responsible for intracellular motility. Its function is to serve as a motor to transport various cargoes on actin filaments. Myo2 has actin binding and cargo binding domains. Mutations in either of these domains cause severe defects in sorting and trafficking at the endosome.”
4.) Lines 188-200: the author simply repeats the same things again and again in these lines and I cannot really see how the length of this part contributes to the manuscript.
Thanks for this comment. We deleted the following fragments containing redundancy: ”The data suggest two alternative pathways leading from monomeric α-synuclein to its inclusion in the Lewy body. The first deposition pathway goes through the intermediate steps of oligomerization and fibril formation. The other includes a dense liquid droplet and a gel-like state rich in amyloid structures before inclusion into the Lewy body. These results are in agreement with the data showing that liquid-liquid phase separation of α-synuclein precedes its aggregation. According to a recent study, in vitro generated α-synuclein liquid-like droplets undergo a liquid-to-solid transition and form an amyloid hydrogel composed of oligomers and fibrillar species.”

Round 2
Reviewer 1 Report
The author is not willing to address my comments adequately.
Reviewer 3 Report
Dear editor,
I read with great interest the revised version of the review article titled “Non-mammalian eukaryotic models untangle the mechanism of neurodegeneration” by Andrei Surguchov. Overall, I think that the author efficiently addressed my concerns and with the additions and modifications the manuscript reads very well. From my side my recommendation is to accept the manuscritpt for publication.